# First Report of Food Poisoning Due to Staphylococcal Enterotoxin Type B in Döner Kebab (Italy)

**DOI:** 10.3390/pathogens12091139

**Published:** 2023-09-06

**Authors:** Angelo Romano, Simona Carrella, Sara Rezza, Yacine Nia, Jacques Antoine Hennekinne, Daniela Manila Bianchi, Francesca Martucci, Fabio Zuccon, Margherita Gulino, Carmela Di Mari, Teresa Zaccaria, Lucia Decastelli

**Affiliations:** 1National Reference Laboratory for Coagulase Positive Staphylococci including S. aureus, Istituto Zooprofilattico Sperimentale del Piemonte Liguria e Valle d’Aosta, 10148 Turin, Italy; 2Laboratory for Food Safety, French Agency for Food, Environmental and Occupational Health & Safety (ANSES), Université Paris-Est, 94700 Maisons-Alfort, France; 3Azienda Sanitaria Locale TO5 di Chieri, Carmagnola, Moncalieri e Nichelino—Dipartimento di Prevenzione SC Igiene degli Alimenti e della Nutrizione, 10042 Nichelino, Italy; 4SC Microbiologia e Virologia Azienda Ospedaliero-Universitaria, Città Della Salute e Della Scienza di Torino, 10126 Turin, Italy

**Keywords:** foodborne poisoning outbreak, staphylococcal enterotoxins, SEB

## Abstract

Staphylococcal food poisoning results from the consumption of food contaminated by staphylococcal enterotoxins. In July 2022, the Turin local health board was notified of a suspected foodborne outbreak involving six children who had consumed döner kebab purchased from a takeaway restaurant. The symptoms (vomiting and nausea) were observed 2–3 h later. A microbiological analysis of the food samples revealed high levels (1.5 × 10^7^ CFU/g) of coagulase-positive staphylococci (CPS). The immunoassay detected a contamination with staphylococcal enterotoxins type B (SEB). The whole genome sequencing of isolates from the food matrix confirmed the staphylococcal enterotoxin genes encoding for type B, which was in line with the SEB detected in the food. This toxin is rarely reported in staphylococcal food poisoning, however, because there is no specific commercial method of detection. The involvement of enterotoxin type P (SEP) was not confirmed, though the corresponding gene (*sep*) was detected in the isolates. Nasal swabs from the restaurant food handlers tested positive for CPS, linking them to the likely source of the food contamination.

## 1. Introduction

*Staphylococcus aureus* is a ubiquitous, opportunistic, and highly versatile pathogen [1,2]. It can survive in hostile environments, colonize the skin and mucous membranes, and cause infection in humans and animals [1,3]. Commonly harbored by approx. 30–60% of healthy people on the skin and in the respiratory mucosa, the toxigenic strains of *S. aureus* can cause a staphylococcal food poisoning outbreak (SFPO) [4] due to incorrect hygiene by food handlers carrying enterotoxin-producing *S. aureus*.

An SFPO can be defined as an intoxication resulting from the consumption of foods containing sufficient amounts of preformed enterotoxins synthesized by *S. aureus* during the logarithmic phase of growth or during the transition from the exponential to the stationary phase [4,5]. The main routes of contamination are via manual contact or respiratory secretions [6]. Food products may be contaminated with *S. aureus* during preparation and processing, which can grow in a wide range of temperatures (7 to 48.5 °C; optimum 30 to 37 °C), pH (4.2 to 9.3; optimum 7 to 7.5), and sodium chloride concentrations up to 15% NaCl [7].

Several types of staphylococcal enterotoxins (SEs) have been reported to cause SFPOs. The most common symptoms are nausea, vomiting, abdominal cramps, and diarrhea 2 to 8 h after the ingestion of the contaminated foods [8]. The condition is usually self-limiting and is usually resolved within 24–48 h after the onset. Occasionally, it can be severe enough to warrant hospitalization, particularly in young children, the elderly, or debilitated people [9].

SEs are low-molecular-weight, single-chain basic globular proteins that share common phylogenetic relationships, structures, functions, and sequence homologies [10]. Based on the similarity of amino acid sequences, Uchiyama et al. [11] divided SEs into three major groups and two minor groups. Group 1 contained the classic SEs such as SEA, SED, SEE and SEO, SEN, SEP, and SEJ. Group 2 contained SEB, SEC, SEG, SER, and SEU. Group 3 was composed of only non-classical SEs such as SEL, SEI, SEK, SEM, and SEQ.

SEs are highly stable to heat treatment, digestive proteinases, irradiation, denaturing agents, and a wide pH range [12]. Only five of the 33 SEs reported in the literature can be detected with commercially available immunoassay kits, namely SEA, SEB, SEC, SED, and SEE [13]. SEA is the most frequently reported SE involved in SFPOs (80%) [14]. However, it has been reported that other SEs have been identified as potential agents of food poisoning, such as SEG, SEH, SEI, SER, SES, and SET [15,16].

SEB has a history of military use as a potential warfare agent [17]. The inhalation of a very small amount (0.004 µg/kg) is effective at inducing symptoms, and a dose of 0.02 µg/kg could be lethal [18]. Due to this potential use, SEB is classified by the U.S. Centers for Disease Control and Prevention (CDC) as a category B agent of potential bioterrorism risk [19]. SEB is a prohibited substance under the Biological Weapons Convention (BWC).

The European Food Safety Authority (EFSA) and the European Centre for Disease Control (ECDC) report that bacterial toxins are the third leading foodborne poisoning agent in the European Union [20]. In 2021 alone, 16 Member States reported 61 foodborne outbreaks classified as “Strong Evidence” (640 cases) caused by staphylococcal toxins [21]. Foods such as meat and meat products, poultry and egg products, milk and dairy products, salads, bakery products, cream-filled pastries and cakes, and sandwich fillings are frequently involved in SFPOs [7]. Staphylococcal toxins have been linked to the highest number of hospitalizations: 51 cases that represent 8% of the cases caused by the causative agent [20].

According to the EFSA data (2010 to 2021), the origin of SFPOs differ widely between countries probably due to differences in food consumption and habits. SFPOs are more often reported to occur in Southern than Northern European countries, which is most likely due to inappropriate storage conditions under critical climate conditions. Le Loir et al. (2003) found that food handlers were the main source of contamination via manual contact with food items or via respiratory droplets from coughing and sneezing [5]. Usually, contamination occurs after the heat treatment of the food. Foods of animal origin, such as raw meat, sausages, raw milk, and raw milk cheese, can be contaminated by animal carriage or infection by *S. aureus*-like mastitis.

Five enterotoxins (SEA–SEE) have been recognized as being responsible for most SFPOs. To our best knowledge, however, no investigation of a foodborne outbreak has been reported based on a complete toolbox of microbiological and immunological methods, not to mention whole genome sequencing. By combining these complementary methods, a SFPO can be better delineated and the source of contamination and the involvement of emerging or newly described enterotoxins can be better identified. 

Five enterotoxins (SEA–SEE) have been recognized as being responsible for most SFPOs. To our best knowledge, however, no investigation of a foodborne outbreak has been reported based on a complete toolbox of microbiological and immunological methods, not to mention whole genome sequencing. By combining these complementary methods, a SFPO can be better delineated and the source of contamination and the involvement of emerging or newly described enterotoxins can be better identified. 

Here, we report an SFPO that occurred in July 2022, involving six children who consumed a takeaway meal of döner kebab. We used a complementary toolbox to identify and confirm the type of enterotoxins involved. The *S. aureus* strains isolated from the food samples and nasal swabs from the food handlers were analyzed to determine the source of the contamination.

## 2. Materials and Methods

### 2.1. Background

In July 2022, the Turin local health board was notified of a suspected foodborne outbreak (FBO) involving six children (three boys and three girls; average age 13 years) who manifested vomiting and nausea 2–3 h after eating. Four were treated at the emergency department. All six had eaten a meal consisting of döner kebab from a takeaway restaurant.

### 2.2. Sample Collection

The food hygiene inspectors from the local health board collected five samples of döner kebabs and four sauces (ketchup, mayonnaise, spicy sauce, and yogurt sauce). The suspicious residual döner kebabs and sauces collected at the restaurant were transported under refrigeration (1–8 °C) to our laboratory where they were analyzed. In addition, nasal swabs were collected from two restaurant food handlers who were involved in preparing the food.

### 2.3. Laboratory Analysis of Food Samples

The food samples were analyzed for the enumeration of CPS, according to standards UNI EN ISO 6888-2: 2021 [22]. Briefly, 10 g of the sampled cheese was added to nine parts of buffered peptone water (90 mL) and mixed. A total of 1 mL was then pour-plated with a freshly prepared rabbit plasma fibrinogen agar medium, following appropriate decimal dilutions and incubated at 37 °C ± 1°C for 24 h. The samples were also tested for staphylococcal enterotoxins according to ISO 19020: 2017 [23]. The first part of this protocol included dialysis concentration. A total of 25 g of the sample was mixed in 40 mL of distilled water at 38 ± 2 °C using rotary homogenizer. The samples were shaken at room temperature (18 °C to 27 °C) for 30 to 60 min to allow for toxin diffusion and the pH of the slurry was adjusted between 3.5 and 4.0 using appropriate hydrochloric acid solutions. The samples were centrifuged at a minimum of 3130× *g* for 15 min under refrigeration (approximatively 4 °C). The aqueous supernatant was sampled and adjusted to pH 7.5 ± 0.1 using sodium hydroxide solutions and centrifuged. The supernatants were filtered through glass wool and concentrated using 50/60 cm of a dialysis membrane with a molecular weight cut-off (MWCO) of 6000–8000 Da (Spectrum Laboratories Inc., Rancho Dominguez, CA, USA) against 30% (*w*/*w*) polyethylene glycol 20,000 (Merck, Darmstadt, Germany) overnight at 4 °C. The concentrated extract was recovered and adjusted to a final weight of 5.25 ± 0.25 g using phosphate buffered saline. The second part of the SEs detection was performed starting from the concentrated extract using a commercially available enzyme-linked fluorescent immunoassay (ELFA), Vidas SET2 (bioMérieux, Marcy l’Etoile, France). That method was used to simultaneously detect SEA to SEE in the food matrices without differentiating the five SEs. The European Reference Laboratory (EURL) for CPS performed confirmatory tests of the samples, which were positive according to ISO 19020: 2017 using an in-house enzyme-linked immunosorbent assay (ELISA) for the detection of SEs type SEA to SEE in all the food matrices [24]. 

In addition, the food samples were analyzed according to UNI ISO 16649-2:2010 for β-glucuronidase-positive *Escherichia coli* [25]. Briefly, every sample was prepared by homogenizing 25 g amounts in 225 mL of a chilled peptone phosphate buffer using a stomacher. Homogenates (0.1 mL) were spotted in duplicate onto the Tryptone Bile X-glucuronide Agar (TBX) (Sigma-Aldrich, St. Louis, MI, USA) then incubated at 44 °C ± 1 °C for 18–24 h. Colonies counts were carried out considering the deep blue β-glucuronidase-positive colonies. The samples were also screened according to ISO 21528-2: 2017 [26] for the detection and enumeration of Enterobacteriaceae. The procedure consisted of the following. The initial suspension and decimal dilutions were prepared from the test sample. The violet red bile glucose (VRBG) agar (Sigma-Aldrich, St. Louis, MI, USA) was inoculated with 0.1 mL of the initial suspension and an overlay of the same medium was added. In addition, decimal dilutions of the initial suspension were prepared under the same conditions and all the dishes were incubated at 37 °C ± 1 °C for 24 h. The colonies of presumptive *Enterobacteriaceae* that could appear as colonies of pink to red or purple (with or without precipitation haloes) were sub-cultured onto the surface of the pre-dried non-selective agar medium and incubated at 37 °C ± 1 °C for 24 h ± 2 h. An isolated colony from each of the incubated plates was selected for the biochemical confirmation tests, such as the oxidase reaction and fermentation glucose test. The colonies that were oxidase-negative and glucose-positive were confirmed as Enterobacteriaceae.

Finally, the food samples were analyzed for *Salmonella* spp. according to AFNOR BRD 07/6–07/04. Briefly, 25 g of every sample were incubated in pre-warmed buffered peptone water (BPW) broth (BIO-RAD, Hercules, CA, USA) for 8/16 h at 41.5 °C ± 1 °C, diluted to 1:10. After enrichment, the DNA extraction was operated by the automated system iQ-Check (BIO-RAD, Hercules, CA, USA) and a real-time PCR was performed on 5 mL of the DNA extract on CFX96 (BIO-RAD, Hercules, CA, USA) using the iQ-Check *Salmonella* II kit (BIO-RAD, Hercules, California, United States).

Our laboratory and all the methods performed in this investigation were accredited according to ISO 17025: 2017 [27].

### 2.4. S. aureus Strain Characterization

For each sample, five colonies displaying the phenotype characteristic for CPS were streaked on blood agar plates and identified as *S. aureus* using the VITEK system GP card (bioMérieux, Craponne, FR). The swab samples were placed in 10 mL of Mueller-Hinton broth (Oxoid, Hampshire, UK) supplemented with 6.5% NaCl (Sigma-Aldrich, Wicklow, Ireland), vortexed, and incubated for 24 h at 37 °C. After incubation, an aliquot of 100 µL was pour-plated onto rabbit plasma fibrinogen Baird-Parker agar plates (Oxoid, Hampshire, UK) and incubated at 37 °C. The species confirmation for the colonies displaying an *S. aureus* characteristic phenotype after 24/48 h was performed using a VITEK system GP card (bioMérieux, Craponne, FR).

Fifteen *S. aureus* strains isolated from the food samples and the nasal swabs from the food handlers underwent a multiplex PCR (mPCR) for detecting the genes encoding SEs (Italian National Reference Laboratory for CPS, Turin, Italy), according to EURL CPS methods [28,29]. The protocols included the detection of the genes from *sea* to *see* and *ser* for the first mPCR and from *seg* to *sej* and *sep* for the second mPCR.

The DNA was extracted starting from single colonies plated on Columbia Blood Agar (Biolife, Milan, IT) for 16–20 h at 37 °C using the Extractme Genomic DNA isolation kit (Blirt, Gdańsk, Poland). A pre-lysis step was added consisting of a treatment of 105 µL of lysozyme (10 mg/mL) and 15 µL of lysostaphin (10 mg/mL) for 30 min at 37 °C. The DNA was quantified using a Qubit Fluorometer (Thermo Fisher Scientific, Waltham, MA, USA). Before the library preparation using the Illumina DNA Library Prep Kit (Illumina, San Diego, CA, USA) following manufacturer’s instructions, the genomes were sequenced using an Illumina MiSeq system (Illumina, San Diego, CA, USA) and MiSeq V3 chemistry in a run 2 × 151 bp paired-end reads.

#### Data Analysis of Whole Genome Sequencing 

All the bioinformatic analyses were performed using the tools on the Staphylococcus *aureus* Intensive Server for Research in Omics data (SIRIO) Galaxy instance (http://90.147.102.165/galaxy accessed on 1 August 2023): an online user-friendly Galaxy interface for performing raw read trimming, assembly, and SE gene detection [30,31]. The raw reads were trimmed using Trimmomatic 0.38 [32] by removing the Nextera adaptors and other Illumina-specific sequences (Illuminaclip set to Nextera (paired-ended)), by removing low-quality residues at the start and the end of the reads (leading:10 and trailing:10), by clipping the reads when the average Q-scores dropped below 20 over a sliding window of four residues (slidingwindow:4:20), and by dropping the reads shorter than 40 bases after processing (minlen:40). The trimmed reads were assembled de novo using Unicycler 0.4.8.0 [33] for the bridging mode moderate contig size and misassembly rate (the bridging mode was set to normal) and the contigs below 200 bp in length were excluded (excluded contigs from the FASTA file which were shorter than this length (bp) set to 200). The relevant assembly statistics (N50, the number of contigs and median coverage against assembly) were calculated using Quast 5.0.2 [34]. 

The assembled genomes were processed using MLST 2.0 (accessed via https://cge.food.dtu.dk/services/MLST/ accessed on 1 August 2022) for the strain type identification using multilocus sequence typing (MLST) [35] and analyzed for the virulence gene detection using VirulenceFinder 2.0 (accessed via https://cge.food.dtu.dk/services/VirulenceFinder/ accessed on 1 August 2022), selecting 90% as the threshold for identification and 60% for the minimum length [36].

Finally, a phylogenic comparison based on single nucleotide polymorphism (SNP) typing was made using CSI Phylogeny 1.2 (accessed via https://cge.cbs.dtu.dk/services/CSIPhylogeny accessed on 1 August 2022) [37], including a non-enterotoxigenic strain as the outgroup. An SNP analysis was performed using the following parameters: 10 × minimum depth at the SNP position, 10% minimum relative depth at the SNP position, 100 bp minimum distance between the SNPs, 30 for the minimum SNP quality, 25 for the minimum read mapping quality, 1.96 minimum Z-score for each SNP, including *S. aureus* NCTC 8325 (GenBank accession number: NC_007795.1) as the reference. The maximum likelihood tree was calculated and visualized using MEGA v.11 [38].

## 3. Results

### 3.1. Sample Collection

Local health department inspectors collected five samples of residual döner kebabs, four sauces (ketchup, mayonnaise, spicy sauce, and yogurt sauce), and nasal swabs from the two food handlers involved in the food preparation.

### 3.2. Laboratory Analysis of the Food Samples

The microbiological analysis of the food samples revealed high levels of coagulase-positive staphylococci (CPS) in all five samples (Table 1). The CPS isolates from each sample were enumerated, characterized, and identified as *S. aureus*. The global qualitative immunoassay (EN ISO 19020) detected SEs (type SEA to SEE) in the food samples. The enzyme-linked immunosorbent assay (ELISA) against SEA, SEB, SEC, and SED in the food matrix returned a positive result only for type SEB.

Table 1 presents the estimated concentration of SEB (range 0.49 ng/g to 1.78 ng/g) in the five food samples. The microbiological analysis revealed β-glucuronidase-positive *Escherichia coli* in the five kebab samples (2.5 × 10^2^ to 3.5 × 10^4^ CFU/g) and in the other food matrices (<10 CFU/g), along with Enterobacteriaceae in the five kebab samples (2.3 × 10^5^ to 3.5 × 10^7^ CFU/g) and in the other food matrices (<10 CFU/g). The tests for *Salmonella* spp. returned negative for all the food samples.

### 3.3. Characterization of S. aureus Strains 

Five CPS isolated from the kebabs and identified as *S. aureus* were characterized. The same characterization was carried out on 10 CPS isolated from the two food handlers after the identification as *S. aureus*. All the strains had the same toxigenic molecular profile harboring *seb* and *sep* genes.

#### Whole Genome Sequencing 

The draft genomes sequence ranged from 2661 Mbp to 2668 Mbp (mean 2664 Mbp) and the GC content ranged from 32.61% to 32.62%. The number of contigs (≥1000 bp) ranged from 18 to 24 (mean 20) and contig N50 ranged from 247.346 to 589.540 (mean 313.175) (Appendix A). All the assembled genomes belonged to the strain type ST-8 (Table 2). The genome data analysis revealed the *seb*, *sep*, *sex*, and *selW* genes encoding SEs or staphylococcal enterotoxins-like (SEls) genes (Table 2).

The distance matrix obtained from the single nucleotide polymorphism (SNP) analysis using CSI Phylogeny 1.2 [37] (Appendix A) and the related maximum likelihood tree created and visualized using MEGA v.11 [38] revealed that the strains were closely correlated (Figure 1).

## 4. Discussion

The rapid onset of illness and the type of symptoms suggested that this foodborne outbreak was caused by bacterial toxins, and that the agents were CPS, including *S. aureus*. Hennekinne *et al.* [39] reported that one or more of the following criteria generally confirm the event of a SFPO: the recovery of more than 10^5^ CFU/g of CPS, the detection of SEs in food leftovers, the isolation of the same *S. aureus* strain both from patient vomitus or stool samples and food leftovers, and/or the identification of *S. aureus* from food leftovers. In the present event, the analysis of the food matrix revealed high counts of CPS and SEB only in the kebabs. Absent all other pathogens, our findings indicated that the SEs were the cause of this SFPO. 

Studies of SFPOs in which the enterotoxigenic strains were isolated from the patients, food, and food handlers are rare [40]. No samples from the patients were collected in the present investigation. However, the molecular characterization of *S. aureus* isolates using the PCR assay and WGS revealed the link between the human carriers and food contamination.

Moreover, the estimated dose of SEs in the food samples ranged from approx. 0.493 ng/g to 1.784 ng/g of SEB, which meant that a range of approx. 49 ng to 170 ng of SEB in the meal was consumed, assuming a 100 g serving size. To our knowledge, the effective dose of SEB that causes poisoning episodes in humans has not been defined. Several SFPOs were characterized with several levels of SEs. The amount of SEA ranked from 1.9 ng to 14.7 ng in potatoes [41] and the SEs amount in pasta was estimated from approx. 8.3 ng to 16.5 ng for SEA and from 13 ng to 26 ng for SED [8]. SEA was the only enterotoxin for which sufficient data were available for dose–response modeling related to food consumption by humans, and the dose that induces effects in 10% of the exposed population was defined as 6.1 ng [42].

Döner kebab is prepared from meat slices (1–6 mm thick), minced meat, and fat (2–4 mm thick) that is marinated for 3 to 6 h. A spice mixture usually composed of white pepper, black pepper, cumin, spice, and thyme is then added. The meat, fat, and minced meat are shaped into a cone [43], the cone is then frozen and sold as is to retail shops. In traditional kebab restaurants, the döner kebab cone is grilled on its outer surface on a vertical rotating spit. When the first outer layer is sufficiently cooked, it is sliced and served alone or together with a variety of sauces and vegetables [44]. The meat can come from different animal species, such as mutton, beef, chicken, or turkey, and include other ingredients such as salt, spices, onions, oil, milk and milk protein, eggs and egg powder, soy protein, and phosphate [45].

Between 2000 and late 2022, the EU Rapid Alert System for Food and Feed (RASFF) reported 41 notifications regarding kebabs. *Salmonella* spp. was the leading cause for notification followed by *Listeria monocytogenes*. However, SEs or enterotoxigenic *S. aureus* were never reported.

The microbiological risks associated with this ethnic meal are primarily the microbiological safety of the raw materials and hygienic working conditions, especially the thawing and cooking procedures. When the outer surface of the kebab cone is exposed to a heat source, the internal part, which thaws during cooking, gradually warms to temperatures ideal for bacterial proliferation.

Furthermore, slicing the meat from the cone and leaving the slices uncovered until sold could also promote the growth of *S. aureus* and increase toxin production because the food is not kept hot by further cooking. Bacterial growth and toxin production may also occur during food product transport at uncontrolled temperatures. This last scenario could fit the SFPO reported here, since all the isolates had the same toxigenic gene profile, and the human carriers were the most probable source of contamination in the outbreak. Previous outbreak investigations have suggested that the improper handling of cooked or processed foods is a major source of contamination, usually due to incorrect temperature control, which allows *S. aureus* to form SEs [7].

The biomolecular biology methods were in line with the results of the ELISA for identifying SEs. All the strains harbored the gene encoding SEB and the same toxin was detected in the food samples, suggesting that the strains produced the toxin. Our findings suggested that characterization of the isolated strains should be applied as a specific tool in SFPO investigations. The detection of the SEs encoding genes in the strains isolated from the matrices allowed us to establish links between the different matrices and identify the potential source of contamination. Furthermore, the data demonstrated the usefulness of the biomolecular methods as complementary tools for investigating foodborne outbreaks caused by enterotoxins when CPS is present in the food.

Moreover, all the isolated strains harbored the same SE gene pattern (*seb, sep, sex*, and *selW*). A recent work demonstrated the production of SEP in brain heart infusion (BHI) broth at a high level [46] and also showed the capability of the *S. aureus* strains isolated in the frame of SFPOs for producing at least two types of SEs in contaminated food [8,14]. Even though SEP has not been reported in SFPOs, based on the data reported by Omoe *et al.,* [16] it could hypothesize the contribution of both toxins in the present SFPO.

WGS enabled the complete profiling of the SE genes in a single assay, without the limitations of the PCR-based methods for detecting genes [47]. The non-classical SEs could not be detected using the existing commercial kits, but they could be responsible for emetic action. In this case, the SFPO remained at a weak evidence step and could not be fully confirmed. However, the complete SE gene profile is essential in SFPO surveillance and in terms of prevention. Moreover, using the WGS approach for whole SE gene profiling of the strains involved in SFPO, more knowledge can be acquired about the SE gene patterns that are most involved in outbreaks.

In the European Union (EU), the Commission Regulation (EC) No 2073/2005 sets out the microbiological criteria for foodstuffs to ensure food safety. Under this regulation, meat and meat products are subject to specific microbiological criteria. In particular, the food safety and process hygiene criteria for meat and meat products principally concern the presence of Salmonella spp, *Listeria monocytogenes*, *E. coli*, and *Enterobacteriaceae*. However, the regulation does not take into consideration either the CPS count or the presence of enterotoxins. Since the regulations and guidelines related to food safety are subject to periodic review in light of new scientific evidence or emerging public health concerns, new amendments will need to highlight how the CPS count could be included among the microbiological criteria for meat products in the near future.

While SEA is the most commonly reported enterotoxin in foods and its predominance in diverse countries has been documented [4], SEB, SEC, or SED alone have also been implicated in SFPOs [48]. To our best knowledge, this is the first SFPO involving kebabs where SEB was found to be the causative agent in Italy. The SEB concentrations observed in this work can be compared to those reported in the literature, where it was stated that SEB resulted in an incapacitating amount of 0.0004 µg/kg [49].

## 5. Conclusions

Our findings in the investigation of this outbreak and the results of microbiological and molecular analyses showed that enterotoxigenic *S. aureus*-producing SEB was responsible for the outbreak. The outbreak resulted from the contamination of food with *S. aureus*, probably from the food handlers via either a skin infection on their bare hands or arms or coughing or sneezing over the food that was not furtherly cooked [50]. Furthermore, this outbreak underlined the rule that food handlers must be properly trained in food hygiene practices in order to prevent the contamination of food and food preparation areas by pathogens [50].

## Figures and Tables

**Figure 1 pathogens-12-01139-f001:**
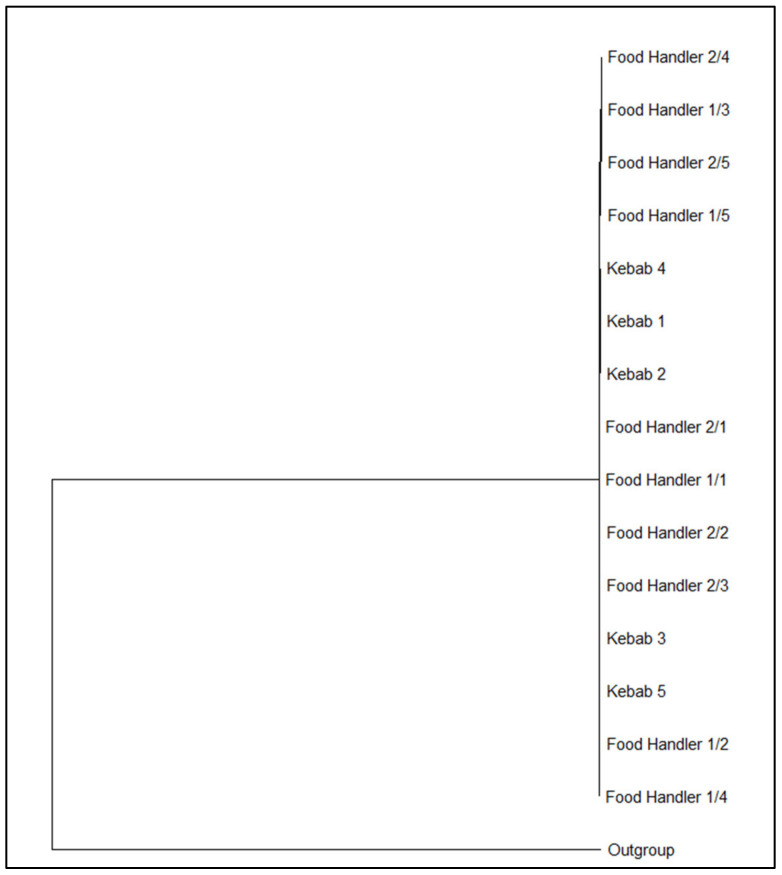
Phylogenetic tree inferred using the maximum likelihood method from the distance matrix based on the single nucleotide polymorphism (SNP) alignment using CSI Phylogeny 1.2 [37] and visualized using MEGA v.11 [38] with 1000 bootstrap replications.

**Table 1 pathogens-12-01139-t001:** Laboratory analysis of the food samples. CFU denotes a colony forming unit; CPS—coagulase-positive staphylococci; n.a. not analyzed.

Food Sample	Sample Unit Number	CPS Enumeration (CFU/g)	Staphylococcal Enterotoxin (Detection)	Staphylococcal Enterotoxins (Specific Identification)
**Döner kebab**	1	>1.5 × 10^7^	Detected	SEB (1.78 ng/g)
	2	>1.5 × 10^7^	Detected	SEB (1.86 ng/g)
	3	>1.5 × 10^7^	Detected	SEB (1.02 ng/g)
	4	>1.5 × 10^7^	Detected	SEB (0.49 ng/g)
	5	>1.5 × 10^7^	Detected	SEB (0.97 ng/g)
**Ketchup**	1	<10	Not Detected	n.a.
**Mayonnaise**	1	<10	Not Detected	n.a.
	2	<10	Not Detected	n.a.
	3	<10	Not Detected	n.a.
	4	<10	Not Detected	n.a.
	5	<10	Not Detected	n.a.
**Spicy sauce**	1	<10	n.a.	n.a.
	2	<10	n.a.	n.a.
	3	<10	n.a.	n.a.
	4	<10	n.a.	n.a.
	5	<10	n.a.	n.a.
**Yogurt sauce**	1	<10	Not Detected	n.a.
	2	<10	Not Detected	n.a.
	3	<10	Not Detected	n.a.
	4	<10	Not Detected	n.a.
	5	<10	Not Detected	n.a.

**Table 2 pathogens-12-01139-t002:** Genome analysis for the multilocus sequence typing (MLST) attribution and virulence gene identification.

Sample		MLST 2.0	VirulenceFinder 2.0
**Food Handler No. 1**	**1/1**	ST-8	*seb, sep, sex, selW, hlgA, hlgB, hlgC, lukD, lukE*
	**1/2**	ST-8	*seb, sep, sex, selW, hlgA, hlgB, hlgC, lukD, lukE*
	**1/3**	ST-8	*seb, sep, sex, selW, hlgA, hlgB, hlgC, lukD, lukE*
	**1/4**	ST-8	*seb, sep, sex, selW, hlgA, hlgB, hlgC, lukD, lukE*
	**1/5**	ST-8	*seb, sep, sex, selW, hlgA, hlgB, hlgC, lukD, lukE*
**Food Handler No. 2**	**2/1**	ST-8	*seb, sep, sex, selW, hlgA, hlgB, hlgC, lukD, lukE*
	**2/2**	ST-8	*seb, sep, sex, selW, hlgA, hlgB, hlgC, lukD, lukE*
	**2/3**	ST-8	*seb, sep, sex, selW, hlgA, hlgB, hlgC, lukD, lukE*
	**2/4**	ST-8	*seb, sep, sex, selW, hlgA, hlgB, hlgC, lukD, lukE*
	**2/5**	ST-8	*seb, sep, sex, selW, hlgA, hlgB, hlgC, lukD, lukE*
**Kebab**	**1**	ST-8	*seb, sep, sex, selW, hlgA, hlgB, hlgC, lukD, lukE*
	**2**	ST-8	*seb, sep, sex, selW, hlgA, hlgB, hlgC, lukD, lukE*
	**3**	ST-8	*seb, sep, sex, selW, hlgA, hlgB, hlgC, lukD, lukE*
	**4**	ST-8	*seb, sep, sex, selW, hlgA, hlgB, hlgC, lukD, lukE*
	**5**	ST-8	*seb, sep, sex, selW, hlgA, hlgB, hlgC, lukD, lukE*

## Data Availability

The datasets used and/or analyzed during the current study are available from the corresponding author upon reasonable request.

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
