# Peer review of "First Report of Food Poisoning Due to Staphylococcal Enterotoxin Type B in Döner Kebab (Italy)"

_pathogens, 2023, doi:10.3390/pathogens12091139_

Round 1
Reviewer 1 Report
Line 187. Should not Staphylococcus be italicized like aureus? Also, checkout line 336 (Salmonella should be, but Enterobacteriaceae should not).
Line 233. Enterobacteriaceae (E capitalized).
Table 1. Option is to remove the data on the ingredients since all were negative.
The was little reviewed on previous donair outbreaks around the world; these should be added.
Author Response
Please see the attacment.

Reviewer 2 Report
Hello, We found this article interesting as a case study without being transcendental. The article makes a valuable contribution to the field of food safety by presenting a method, not really innovative but nevertheless effective, for the detection of staphylococcal enterotoxins (SEA, SEB, ...... ), which are generally secreted into food during meal preparation and remain a major cause of human foodborne contamination. Here, the authors detected and identified the presence of SEB in kebabs, a food product widely consumed in Italy. In-depth analysis of the kebab samples collected revealed the presence of toxins hazardous to human health in all their finished product samples, without being able to attribute their origin to the different ingredients that make up the kebabs. Whole genome sequencing of the isolates from the food matrix confirmed the presence of staphylococcal enterotoxin genes coding for type B, corresponding to the SEB found in the food. In addition, nasal swabs from the restaurant food handlers were positive, linking them to the likely source of food contamination.
- Contrary to what the authors state regarding the absence of a specific detection method, it seems to me that very similar studies have been carried out on SEB using technologies very similar to the classic ELISA and in different food matrices (milk, cake, cheese and chicken), such as Detection of Staphylococcus enterotoxin B (SEB) using an immunochromatographic test strip. Mehrdad G et al. Isolation and identification of enterotoxigenic Staphylococcus aureus isolates from Indian food samples: evaluation of an in-house developed aptamer linked sandwich ELISA (ALISA) method. Sundararaj N et al. Finally, SEB has been widely detected in S. argenteus isolates (100% homology) by ELISA by several groups (A Sensitive Immunodetection Assay Using Antibodies Specific to Staphylococcal Enterotoxin B Produced by Baculovirus Expression, Ju-Hong Jang et al; First report of enterotoxigenic Staphylococcus argenteus as a foodborne pathogen, Cavaiuolo M. et al. This is a non-exhaustive list.
With regard to the state of the art, the authors could complete their references by referring to previous publications on the detection and determination of SEB in other food matrices.
In short, the work is of high quality, with a good application in terms of detection and identification, without being extraordinary.
Yours sincerely.
Reviewer 3 Report
The manuscript by Romano et al., is well written reporting staphylococcal food poisoning due to staphylococcal enterotoxin type B in doner kebab. However, its better to revise the presentation of manuscript.
1) Line 52 to Line 62 can be presented in the table to get the better picture for the readers.
2) If possible, its better to present Result shown in Table 1 in the bar graphs as a figure.
Reviewer 4 Report
Dear Authors,
The paper sounds good in terms of presentation. However I have two concerns regarding this paper. First one is where are the accession numbers of the whole genome sequences presented in the phylogenetic tree. Moreover why the specimens from the children are not compared in this tree.
English quality is fine.
Reviewer 5 Report
Reviewer comments
Manuscript number: pathogens-2579030
Title: First report of staphylococcal food poisoning due to staphylococcal enterotoxin type B in döner kebab (Italy)
General comment: This study aims to report an SFPO that occurred in July 2022 involving six children who consumed a takeaway meal of döner kebab and determine the source of contamination. This study will be of interest to the readers.
Detailed comments:
Title: Suggest author check if it is okey to remove “staphylococcal” in “staphylococcal food poisoning”, since you have mentioned in the “staphylococcal enterotoxin type B…”, and state “in Italy” or “in Turin, Italy”.
“First report of food poisoning due to staphylococcal enterotoxin type B in döner kebab in Italy.
Abstract:
Introduction:
Line 69-85: beside S. aureus, gene, Can author elaborate a bit more of information about döner kebab process/prepare/ready to eat food, and S. aureus/CPS contamination situation on that food (if any relevant in previous studies, reports elsewhere), even you had some in lines 282-294!
Methods:
Line 107 (and line 160): which laboratory, can you define? or delete “our”
Line 103-4: can author describe a bit more about döner kebab samples: including what?
Section 2.3. Line 165-8: for nasal swab sample analysis can be moved to section 2.2 and can separate food and human sample?
No further comments on other parts/sections!
